# Spider-silk inspired polymeric networks by harnessing the mechanical potential of β-sheets through network guided assembly

Nicholas Jun-An Chan [1,3], Dunyin Gu[1,3], Shereen Tan[1], Qiang Fu[1], Thomas Geoffrey Pattison [1], Andrea J. O'Connor [2] & Greg G. Qiao [1✉]

The high toughness of natural spider-silk is attributed to their unique β-sheet secondary structures. However, the preparation of mechanically strong β-sheet rich materials remains a significant challenge due to challenges involved in processing the polymers/proteins, and managing the assembly of the hydrophobic residues. Inspired by spider-silk, our approach effectively utilizes the superior mechanical toughness and stability afforded by localised β-sheet domains within an amorphous network. Using a grafting-from polymerisation approach within an amorphous hydrophilic network allows for spatially controlled growth of poly (valine) and poly(valine-r-glycine) as β-sheet forming polypeptides via N-carboxyanhydride ring opening polymerisation. The resulting continuous β-sheet nanocrystal network exhibits improved compressive strength and stiffness over the initial network lacking β-sheets of up to 30 MPa (300 times greater than the initial network) and 6 MPa (100 times greater than the initial network) respectively. The network demonstrates improved resistance to strong acid, base and protein denaturants over 28 days.

[1] Polymer Science Group, Department of Chemical Engineering, University of Melbourne, Parkville, Melbourne, VIC 3010, Australia. [2] Department of Biomedical Engineering, University of Melbourne, Parkville, Melbourne, VIC 3010, Australia. [3] These authors contributed equally: Nicholas Jun-An Chan, Dunyin Gu. ✉email: gregghq@unimelb.edu.au

Polypeptides are the fundamental building blocks of many naturally occurring materials due to their ability to fold into specific yet complex molecular architectures, including α-helices and β-sheets[1–4]. The ability to form these secondary structures in response to various environmental cues (e.g. protein concentration[5,6], metallic ion concentration[7,8] and pH[9]), as well as their diverse functionality, has made them a cornerstone of biochemical research. The importance of how these secondary structures affect material properties is exemplified in naturally occurring dragline spider-silk; a natural polypeptide-based structure which displays a high tensile strength comparable to high tensile steel[10]. Specifically, its excellent mechanical properties are attributed to the spatial arrangement of the amino acids within the polypeptide, which arrange to form higher-order β-sheet architectures through hydrogen bonding primarily from the hydrophobic amino acid residues (i.e. alanine)[11–14]. Surrounding these β-sheet architectures is an arrangement of semi-amorphous, highly extendable glycine-rich regions. This specific arrangement (i.e. composition and spatial) leads to spider silk displaying incredible toughness—a tensile strength between 0.88–1.5 GPa coupled with an extension at break of 21–27 %[10,13]. This incredible mechanical potential garners significant interest towards the utilisation of polypeptides in synthetic materials, such as hydrogels, films and fibres[15–19] for a wide range of applications, including tissue engineering and drug delivery[20–22].

When synthesising β-sheet forming peptides, both the synthetic technique and the molecular components need to be carefully considered. Synthesis via genetically modified microbes and solid-state synthesis are both common methods for producing peptides, but can require acute knowledge of the peptide sequences that the genes are derived from in the case of the former, and in the case of the latter, solid-state synthesis is limited to the fabrication short chain peptides due to decreasing reaction yield with increased coupling cycles for each amino acid[23–29]. N-Carboxyanhydride ring-opening polymerization (NCA ROP) is comparatively a cheaper and a more simple method towards the synthesis of large quantities of polypeptides with a greater potential to create long chain polypeptides, although the method lacks the same level of sequence control observed in solid-state synthesis and genetic engineering[30–35]. Despite this lack of control, we have recently shown the star-shaped polymers synthesised from NCA ROP 'arms' are able to observe equivalent, if not better anti-microbial activity when compared to sequence-controlled polypeptides (anti-microbial peptides (AMPs))[36]. Specifically, when designing polypeptides for β-sheet formation, the control of hydrophobic association is fundamental, yet difficult to achieve in the laboratory. This generally leads to ill-defined, uncontrolled and poorly soluble aggregates[14,37], which greatly impacts the mechanical potential of the formed structures (e.g. bulk hydrogels)[14,37].

Generally, attempts commonly used to allay the uncontrolled association of hydrophobic residues and subsequent unusable aggregates introduce a high ratio of hydrophilic components to form polypeptides[22]. A grafting-to approach may then be used to subsequently incorporate peptides into a hydrogel[38,39]. For example, Clarke, Pashuck[40] have reported the use of grafted β-sheet forming peptide sequences to the polymeric backbone of hydrogels to induce self-healing properties, however because the peptides remained short with a high hydrophilic residue content, mechanical performance was limited. When employing NCA ROP to synthesise linear polypeptides, hydrophilic blocks such as poly(ethylene glycol)[34,35] or hydrophilic polypeptides[31–33] have often been used to prevent this uncontrolled association from occurring, but their inclusion is highly detrimental to the polypeptides mechanical potential due to the disruption of the hydrophobic association[31–35]. Natural spider-silk uses blocks of alanine to form β-sheets during the natural silk fiber production process which involves specific environmental changes to encourage the formation of the β-sheets[10]. However, it should be noted that although it is well known that alanine forms β-sheets at low molecular weight (small blocks), high molecular weight (larger blocks) alanine has been shown to form α-helices[41–43]. Our group has had previous success using valine NCA ROP to form β-sheets, making valine a viable option for constructing spider-silk inspired materials[44].

Herein, we report a simple method to mimic spider-silk by creating localised β-sheet domains, which endow the formed bulk hydrogels with superior mechanical performance. A grafting-from approach is used to incorporate β-sheet forming polypeptides into a pre-fabricated three-dimensional hydrophilic network, which acts as a template to guide β-sheet assembly to mimic the amorphous matrix found in spider-silk. Free amine groups embedded in the networks are subsequently used as a site for NCA ROP of valine and glycine resulting in the spatial and controlled formation of crystalline β-sheet nanocrystals. The physical characteristics of the β-sheet reinforced networks are reported in terms of compressive characteristics, revealing a three-orders of magnitude increase in toughness attributed to the overall β-sheet network rather than the general effect of increased hydrophobicity. Given the natural ease and versatility of polypeptide synthesis via NCA ROP, this synthetic approach can facilitate the incorporation of β-sheets in a far broader range of materials moving into the foreseeable future.

## Results

### β-sheet rich networks synthesis via a grafting-from approach.

To demonstrate a proof of concept, an initial network was synthesised by free radical polymerization with pendant amine groups followed by directed β-sheet incorporation via NCA ROP (Fig. 1). The initial network was designed with pendant amine groups randomly located on the network backbone spaced out with a PEG-based spacer monomer, thus controlling the spatial arrangement of the β-sheet forming polypeptides. The amine groups acted as initiating sites for NCA ROP of β-sheet forming polypeptides, resulting in a polymeric network incorporating non-covalent β-sheet crosslinks. Both hydrogels and cryogels were formed to compare the differences between a conventional (non-porous) and macroporous network. Poly(L-valine) (pVal) chains are known to result in β-sheet formation, resulting in crystalline and rigid β-sheet regions being observed[45]. Thus, glycine was introduced to act as a β-sheet destabilizing residue[12] and being responsible for amorphous β-sheet regions in spider-silk[10]. Thus, L-valine NCA (Val NCA) and glycine NCA (Gly NCA) were synthesised and then used to form crystalline and amorphous β-sheet regions respectively.

The polymerization of β-sheet forming polypeptides via NCA ROP is grown from the available amine groups of the polymer backbone as initiating sites. Hence, Val NCA monomer concentration was varied from 30 to 360 mg/mL to yield increasing pVal chain lengths (Table 1). As expected, the chain length of the valine increased with increasing monomer concentration along with a decrease in conversion. The latter is likely due to the following two reasons. Over the course of β-sheet incorporation, the hydrogels and cryogels noticeably shrink in volume, indicating a reduction in overall pore size which would reduce the diffusion of NCAs into the gels over the course of the reaction. Furthermore, due to the increasing hydrophobicity of networks, the amine groups are hydrophobically shielded from solvent in the late stages of polymerization.

In addition to the valine homopolypeptides formed, hydrogel networks with random copolypeptides were also synthesised with Gly NCA. As glycine is a strong β-sheet breaker[12], it is used to

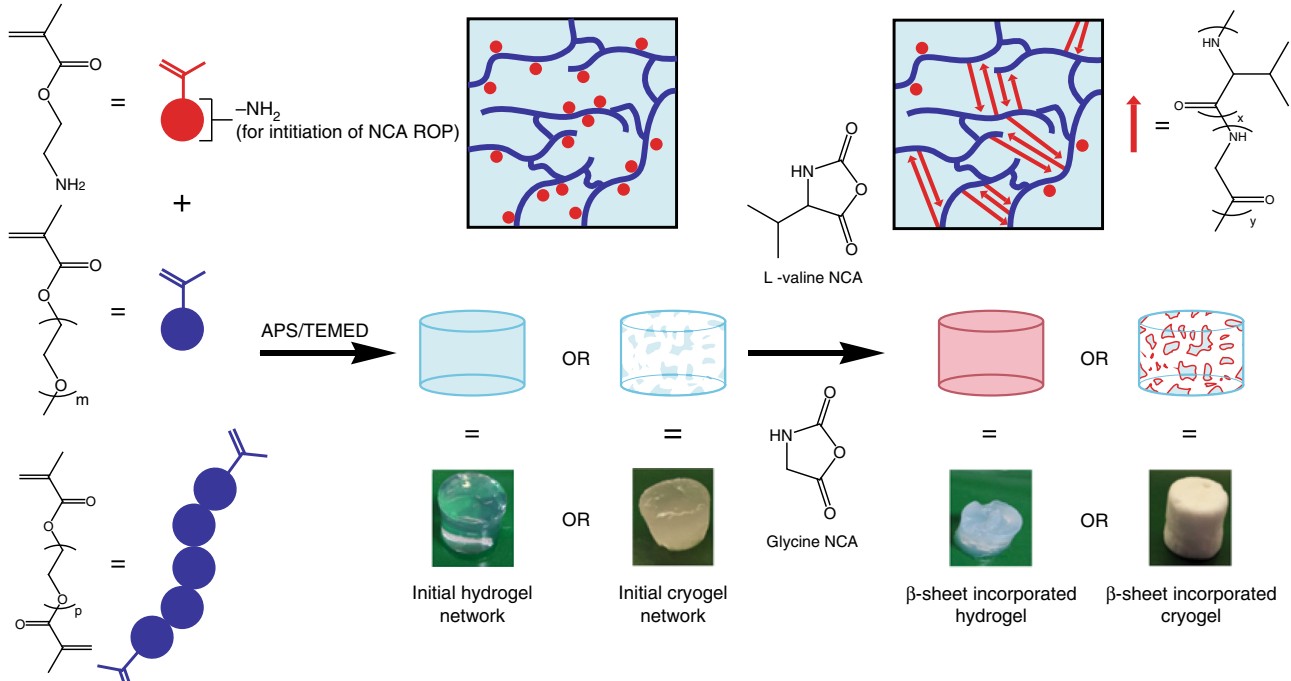

**Fig. 1 Fabrication of peptide β–sheets loaded composite networks.** Networks synthesised via (i) production of two initial networks (a hydrogel and a cryogel) with pendant primary amine groups by free radical polymerization; and (ii) subsequent ROP of Val NCA initiated by the amine groups on the polymer backbone, resulting in the formation of β-sheet secondary structure within the network.

replace small quantities of Val NCA at 5 and 10 mol % within a constant total molar concentration at 2.51 mmol/mL (equivalent molar monomer concentration to HB360 and CB360) in a solution (Table 1). The glycine allowed for slightly longer polypeptides to be formed due to the reduced hydrophobicity associated with the less hydrophobic glycine residue. The conversion of glycine was close to completion while valine conversion was increased to ~82–84%. The amine groups remain hydrophobically shielded from solvent with the low glycine fraction.

**Impact of β-sheet incorporation on network swellability.** The swelling characteristics of the hybrid networks further supports the hydrophobic shielding effect and reduced pore size in β-sheet formation. As expected, the equilibrium swell ratio decreases with increase in the polymerization degree of Val NCA for both hydrogels and cryogels (Fig. 2a). The decreased swelling capacities and subsequent porosity stem from the introduction of hydrophobic pVal grafts and β-sheet conformations. If porosity was the only factor affecting the overall conversion, then polypeptide chain growth would be halted when the decrease of swell ratio stopped. However as evidenced by the cryogels beyond a certain Val NCA concentration (120 mg/mL), the β-sheets-rich cryogels became non-swellable while the conversion continued to drop. The flotation of the cryogels due to a combination of low densities (<750 mg/mL$^3$) and insufficient swellability to take on enough overall mass in water (Fig. 2b, c) further shows evidence of hydrophobicity the non-swellable cryogels remains more buoyant than the surrounding water.

**Morphological analysis of continuous β-sheet linked network.** The morphological impact of β-sheet formation was visualised to ascertain the nature of β-sheet crosslinking and the resulting network. Morphological examination via scanning electron microscopy (SEM) was performed for both hydrogels and cryogels. Examination of the initial networks showed smooth surfaces for

both hydrogels and cryogels as expected with highly swellable polymeric networks (Supplementary Fig. 1). At low levels of β-sheet introduction, β-sheet aggregates could be observed, with the gel surfaces being relatively rough (Fig. 3a, c). When β-sheet content was increased, the aggregates in the hydrogels associated with each other to become a continuous network (Fig. 3b). It was found that β-sheets aggregation was reduced with the introduction of glycine compared to samples soaked in the same molar concentration of NCA and samples soaked in the same amount of valine (Supplementary Fig. 2). The slightly porous structure does indicate the attempted aggregation of β-sheets is occurring, but being interrupted by the glycine, with spherical like bumps appearing on the gel surface, suggesting the presence of more amorphous β-sheet regions. Compared to the network with glycine but no valine, no aggregation is present. The characteristic interconnected pore network of the cryogels (Fig. 3d) is preserved, with β-sheets acting as a hydrophobic layer. The β-sheet aggregates bundle with increasing β-sheet content, though aggregates remain more spherical than the continuous network found in the hydrogels.

The morphology of the β-sheet structure was further assessed using a Thioflavin T (ThT) binding assay under fluorescence microscopy[46,47]. As shown in the confocal image (Fig. 4a), the labelled amyloid clusters are scattered further illustrating the lack of amorphous regions and the local aggregation of the β-sheets being responsible for the aggregates observed in the SEM imaging. Increasing pVal content allowed expansion of β-sheets with lateral aggregation extending (Fig. 4b), resulting in the more continuous network observed in the SEM imaging. The introduction of glycine at low concentrations does still result in crystalline β-sheet structures (Fig. 4c). However, the morphology within these crystalline regions is slightly more amorphous, with less defined structures being visible compared to those with homopoly(L-valine) (Fig. 4d). In the case of β-sheets-rich cryogel, the β-sheet regions are spread out across the porous network walls (Fig. 4e). At closer magnification (Fig. 4f), the β-sheet regions being far more densely packed than the hydrogel.

**Table 1 Estimated polymerization degree of Val/Gly NCA in the initial networks.**

| Experimental Code | Percentage of glycine (mol %) | Val NCA Conc. (mg/mL, mmol/mL) | Gly NCA Conc. (mg/mL, mmol/mL) | Type of initial network | Val Conv.[a] (%) | Gly Conv.[a] (%) | Average DP[b] (Val)[b] | Average DP[b] (Gly)[b] |
|---|---|---|---|---|---|---|---|---|
| HB30 | 0 | 30, 0.21 | N/A | Hydrogel | 90.4 ± 4.8 | N/A | 7.3 ± 0.03 | N/A |
| CB30 | 0 | 30, 0.21 | N/A | Cryogel | 88.8 ± 4.5 | N/A | 7.2 ± 0.03 | N/A |
| HB60 | 0 | 60, 0.42 | N/A | Hydrogel | 86.4 ± 2.3 | N/A | 13.9 ± 0.03 | N/A |
| CB60 | 0 | 60, 0.42 | N/A | Cryogel | 83.2 ± 4.8 | N/A | 13.4 ± 0.06 | N/A |
| HB120 | 0 | 120, 0.83 | N/A | Hydrogel | 85.6 ± 5.5 | N/A | 27.6 ± 0.14 | N/A |
| CB120 | 0 | 120, 0.83 | N/A | Cryogel | 77.8 ± 8.7 | N/A | 25.1 ± 0.22 | N/A |
| HB240 | 0 | 240, 1.67 | N/A | Hydrogel | 80.5 ± 2.5 | N/A | 51.9 ± 0.12 | N/A |
| CB240 | 0 | 240, 1.67 | N/A | Cryogel | 77.8 ± 7.2 | N/A | 50.2 ± 0.36 | N/A |
| HB360 | 0 | 360, 2.51 | N/A | Hydrogel | 75.2 ± 2.0 | N/A | 72.7 ± 0.15 | N/A |
| CB360 | 0 | 360, 2.51 | N/A | Cryogel | 76.8 ± 0.4 | N/A | 74.3 ± 0.03 | N/A |
| GH5 | 5 | 342, 2.39 | 12.7, 0.12 | Hydrogel | 83.6 ± 10.7 | 94.8 ± 3.8 | 76.8 ± 0.77 | 4.6 ± 0.01 |
| GH10 | 10 | 324, 2.26 | 25.4, 0.25 | Hydrogel | 82.6 ± 13.9 | 97.9 ± 0.8 | 71.9 ± 0.94 | 9.5 ± 0.01 |

[a]The NCA monomer conversion rate was determined by $^1$H NMR using a calibration curve, according to the integral ratio between remaining NCA monomer and solvent characteristic peaks stipulated in Methods. ±errors based on standard deviation (n = 3).
[b]Average DP is the average number of grafted Val repeat units per -NH$_2$ group along the network backbone, estimated by the molar ratio between converted Val NCA and -NH$_2$ groups in the initial networks (assuming the initiating efficiency and polymerization rate are the same throughout the networks).

**Crystallinity and parallel/antiparallel nature of β-sheets.** The specific structures and crystallinity were analysed via Attenuated Total Reflectance Fourier Transform Infrared (ATR-FTIR) spectroscopy and X-ray Diffraction (XRD). The FTIR spectra (Fig. 5a) shows a representative range of the networks in this study, with analysis focusing on the numerous peaks assigned to the amide bond. As NCA ROP polymerisation proceeds, the formation of amide functionality is observed with sharp amide A and B bands emerging at around 3280 and 3085 cm$^{-1}$ (due to N–H vibrational stretching), indicating successful polymerization[48]. The amide A band is shown to have a small shoulder at 3300 cm$^{-1}$ once glycine is introduced, which is more evident in samples containing greater amounts of glycine (Supplementary Fig. 3)[48]. As the amide A bond is largely sensitive to hydrogen bonding strength, this can be associated to the change in bond strength. The primary amine assigned to a broad peak between 3300 and 3500 cm$^{-1}$ is clearly seen in the initial network and remains as a shoulder overlapping with the amide A peak in subsequent networks. The presence of this signal is to be expected as the terminal primary amine of polypeptides remains after NCA ROP, though its relative strength is drastically reduced with increasing β-sheet induction due to the increase of amide bonds. The amide I band (C = O stretch vibration) has been identified as the most sensitive spectral region corresponding to different polypeptide secondary structures and used to determine their conformations[49]. Hereby, the strong amide peak around 1627 cm$^{-1}$ with a weak shoulder at 1690 cm$^{-1}$ suggests the formation of β-sheet secondary structure adopted in the mixture of antiparallel and parallel conformations, respectively[49]. The shoulder assigned to the parallel conformations slightly increases with increasing glycine content (most evident in Supplementary Fig. 3), which is known to be less stable than the antiparallel conformation, thus affecting both mechanical and degradation properties[49]. This increase in parallel β-sheets can also be attributed to the lower degree of crystalline β-sheet aggregation observed in the SEM imaging and ThT staining. The amide V band provides more evidence of β-sheet conformations being located around 710 cm$^{-1}$ instead of 610–620 cm$^{-1}$ assigned to α–helices[50]. It should be noted that a broad band with a peak at around 590 cm$^{-1}$ becomes present in the hydrogels containing glycine but is identified instead as an amide VI (C = O out-of-plane bending) as the amide V bond is not diminished in the glycine containing gels[51]. Other distinguishing features of the β-sheet incorporated networks include amide II band (N–H in-plane bending) near 1530 cm$^{-1}$, amide V band (N–H out-of-plane bending) around 710 cm$^{-1}$, and the pVal aliphatic side-chain C–H stretch peaks around 2965 cm$^{-1}$ (which is diminished slightly one glycine is introduced)[48–50]. Also, the characteristic peaks of the initial network are still clearly present in the spectra of β–sheets-incorporated networks, including C–H stretch on the polymer backbone, C = O stretch of ester group and C–O–C stretch in the side chain. The increase in β-sheet content with increasing Val NCA concentrations in the ROP process was also highlighted with the intensity of the β-sheet band (amide I) relative to the polymer backbone stretch in the initial network (C–O–C stretch) (Table S1). The higher intensity ratio (referring to a higher β-sheet content) for the cryogel compared with its hydrogel counterpart is presumably caused by the higher propensity of pVal chains for β-sheets formation in a much more confined space (i.e. highly concentrated pore walls). The formation of β-sheet secondary structure was further verified by X-ray diffraction (XRD) (Fig. 5b). XRD patterns only show a broad peak for the initial network, indicating its amorphous nature, whereas two evident diffraction peaks at 2θ = 9° and 19° can be seen for β-sheets-incorporated networks, attributed to the β-sheet crystalline plane with a typical *d* spacing of 10.1 and

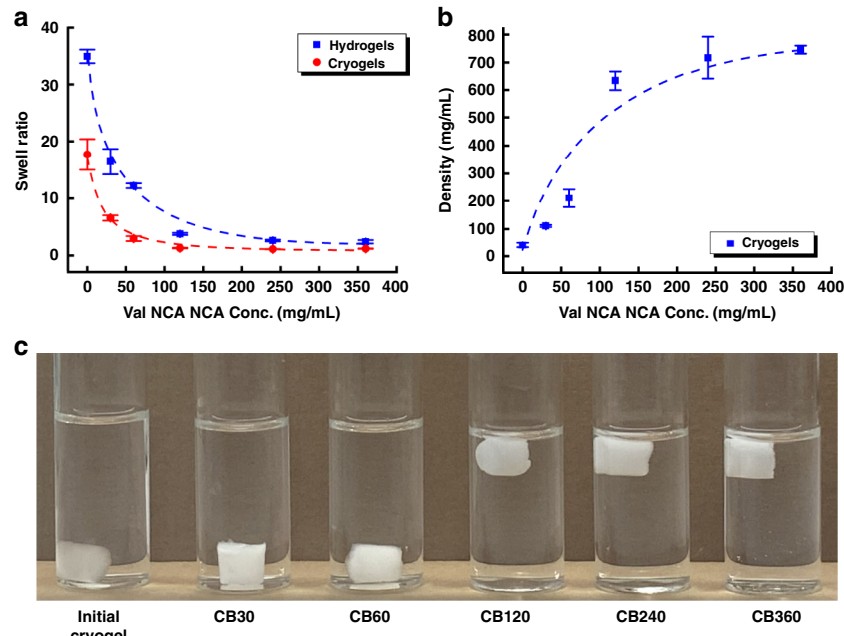

**Fig. 2 Characteristics of networks in deionised water (glass vial diameter ~25 mm) at pH 6.7. a** Equilibrium swell ratios of β-sheets-incorporated hydrogels and cryogels prepared with different Val NCA monomer concentrations (0~360 mg/mL); **b** Summary of bulk density of dry β-sheets-incorporated cryogels as a function of Val NCA concentrations (noting at higher than 120 mg/mL of Val NCA, the cryogels are non-swellable); **c** Photos showing the cryogels with different content of β-sheets. All error bars based on standard deviation ($n = 3$).

4.6 Å, respectively[34,52]. Like the trend observed for FTIR spectra, the two peaks intensify upon elevating the Val NCA concentrations. The transition from broad to sharp peaks suggests the embedment of crystalline β-sheet domains in a semi-amorphous matrix.

**Mechanical toughening impact of β-sheets.** The mechanical toughening via β-sheet incorporation into the networks was characterised by uniaxial compressive testing, observing network behaviour until complete pore disintegration. The easy handling of these gels for compressive mechanical testing sets them distinct from many weak hydrogels physically cross-linked through β-sheets assembly, for which oscillatory rheology was the only applicable mechanical test[31–35]. The stress-strain curves obtained indicate hydrogel stiffening and strength consistently increased with increasing pVal formed β-sheet content with the Young's Modulus increasing by three orders of magnitude (from 2 kPa to 9.4 MPa) (Fig. 6a, b). These features can be attributed to the crystalline β-sheet aggregates identified before, providing a similar property to the double network effect, which is attributed to stiff a second cross-linked network in a soft chemically cross-linked first network. It should be noted that while the effect is similar, the covalent bonding of the β-sheets disqualifies it from being a true double network despite carrying similar properties. The increase in strength and toughness (Fig. 6e) provides further evidence of increased mechanical properties via this methodology. The β-sheet nanocrystals observed in these composite networks by ThT staining and SEM imaging typically feature catastrophic fracture subject to mechanical perturbation, triggered by rupturing of hydrogen bonds that bind β-strands in clusters, as previously reported.

The introduction of glycine was designed to disrupt these β-sheet nanocrystals and help to introduce more amorphous β-sheet regions allowing for a reduction in the brittle nature of the material while minimising strength loss. Toughness remains around 2 MJ/m$^3$, implying that the overall integrity of the gel remains intact despite the introduction of β-sheet forming residues. However, a loss in strength and subsequent increase in compressibility defines the tunable nature of these materials (Fig. 6f), maintaining a similar characteristic curve to the hydrogels with the highest density of β-sheets.

The composite cryogels managed to yield less stiff gels, but with superior strength and compressibility. Cryogels are known to have an intrinsic advantage of being more compressible and mechanically stable over conventional non-porous hydrogels and the weak porous networks prepared otherwise, owing to the densification of polymers in the pore walls. As observed in the ThT staining and SEM imaging, the β-sheets reinforce the existing network pore walls rather than creating completely new walls across pores. The effect of this different morphological reinforcement can be observed when comparing the trends in stiffness in the hydrogel and cryogel morphologies (Fig. 6b, d) with the Young's Modulus of the cryogels only increasing by 2 orders of magnitude (28 kPa to 8 MPa) compared to the 3 orders of magnitude of the hydrogels. Furthermore, the lack of change in the porous network itself is also reflected in the comparative lack of drastic changes in ultimate strain (Fig. 6c), with the initial cryogel having a compressibility of 80%, the most compressible cryogel (due to soaking in 60 mg/mL Val NCA solution) having a compressibility of 88% and the cryogel with the highest β-sheet content having a compressibility of 81%. As expected, the mechanical strength significantly rose by approximately 300 times (from 0.1 to 300 MPa) with 12-fold increase in the Val NCA concentration (from 30 to 360 mg/mL). Further confirmation of the maintenance of the porous network is the distinctive compressive patterns of cellular monoliths (e.g. foams and cork), with three regimes of deformation (inset of Fig. 6c): nearly linear elastic regime, corresponding to bending of pore walls; followed by a near-plateau regime of flat stress, reflecting the easy deformation due to the cell collapse under pressure; and finally an abrupt increase in the stress in the third regime, because of densification of cellular walls.

To ensure that the mechanical enhancement was primarily due β-sheet formation and not simply hydrophobic modification by

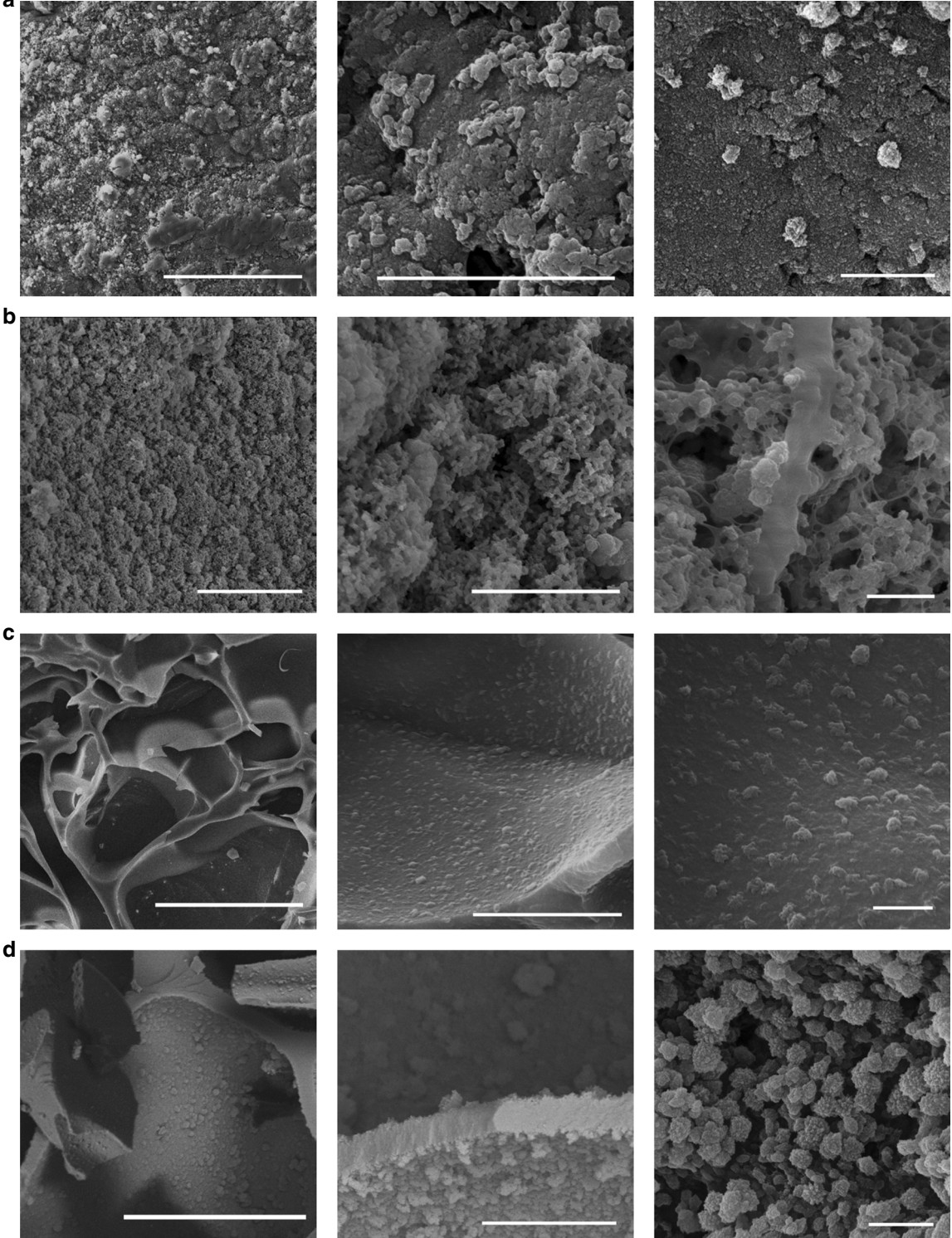

**Fig. 3 SEM cross-sectional images of β-sheets-incorporated hydrogel. a**, **b** Hydrogels and **c**, **d** cryogels with different content of β-sheets, CB60 (**a**, **c**) and CB360 (**b**, **d**). From left to right, scale bars are 50 μm, 5 μm and 1 μm with increasing magnification.

ʟ-valine and glycine as has been seen in other hydrogels, a network grafted with poly(methyl methacrylate) (PMMA), a typical hydrophobic polymer that is hard to crystallize, was synthesised using RAFT (Supplementary Fig. 4)[53–55]. With insertion of more PMMA grafts in the networks, the hydrogels and cryogels became harder and, in the case of the hydrogels, more brittle (Supplementary Fig. 5), which is in agreement with the observations of

hydrophobically modified hydrogels reported in the literature. The stiffness of the β-sheet incorporated gels was approximately 1 order of magnitude higher than the PMMA at the same monomer concentrations in both the hydrogels and the cryogels, where the difference in stiffness become more pronounced with greater monomer concentrations (Supplementary Fig. 5b, d). This result provides a clear indication that the extraordinary mechanical

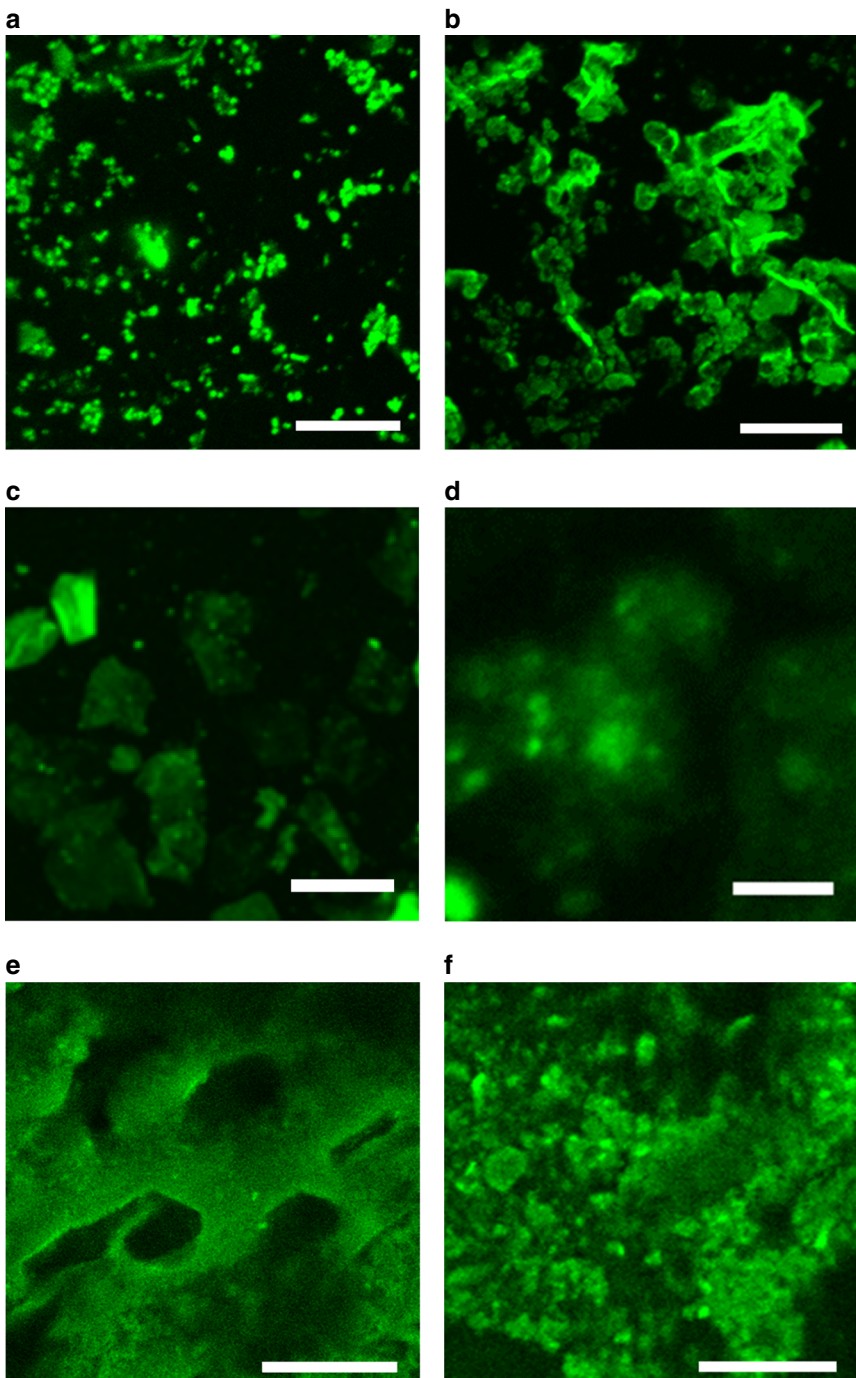

**Fig. 4 Confocal images of β-sheets-incorporated networks stained via ThT assay. a**, **b** Hydrogels with different content of β-sheets from HB60 and HB360, respectively (scale bars are 5 μm); **c** Hydrogel with disrupted β-sheets within GH10 (scale bar is 50 μm); **d** Magnified β-sheets from GH10 (scale bar is 5 μm); **e** Cryogel with β-sheet from CB360 (scale bar is 50 μm); **f** Magnified β-sheets from CB360 (scale bar is 10 μm).

properties, particularly enhanced stiffness, are primarily driven by the rigid self-assembled β-sheet conformations, rather than the hydrophobicity of polypeptide grafts.

**Increased stability of networks under extreme conditions**. In addition to the improved structural integrity, the incorporation of β-sheets confers greater thermal stability to these networks. Thermogravimetric analysis (TGA) (Fig. 7) shows that with more intense β-sheet conformations, the networks (both hydrogels and cryogels) are more thermally stable, as characterised by higher

remaining fraction between 200 and 300 °C. Opposing energetic effects of enthalpically driven hydrogen bonding of β-sheets and entropically driven hydrophobic interactions of the isopropyl side-chain of Val residues are believed to contribute to such superb resistance to elevated thermal treatment[4,56]. All the networks have an onset temperature of mass loss between 150 and 200 °C, corresponding to the disintegration of initial networks. In general, the networks with more β-sheets start to degrade at a higher temperature, with the hydrogels synthesised from 360 mg/mL Val NCA solution displaying significant degradation at 160 °C greater than the initial network and cryogels synthesised

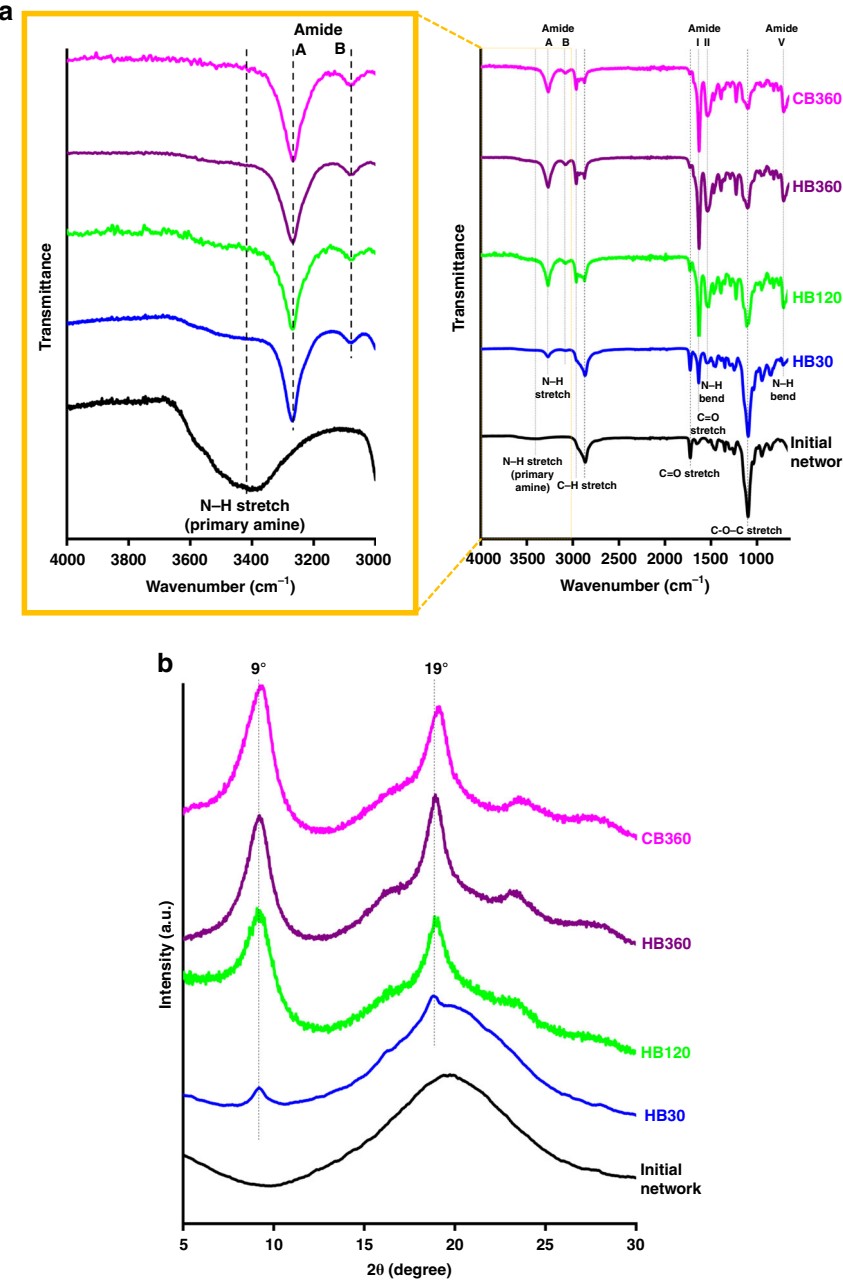

**Fig. 5 Structural characterizations of initial and β-sheets-incorporated networks.** Networks obtained using different Val NCA monomer concentrations (HB30, HB60, HB360, CB360) indicated as the suffix of each legend. **a** ATR-FTIR with a focus on the spectrum between the 3000 cm$^{-1}$ and 4000 cm$^{-1}$; **b** XRD spectra.

from 360 mg/mL Val NCA solution displaying significant degradation at 40 °C greater than the initial network. A thermal degradation process atypical for hydrophobically modified networks was observed with two distinct stages, implying the hybrid structure with different phases[57]. In the wake of a slow mass drop, a much steeper mass loss occurred to β-sheet-incorporated networks from 300 to 400 °C, due to the progressive decomposition of β-sheet crystals in this temperature range[58]. Once the β-sheet network is completely degraded, the β-sheet-incorporated networks follow a similar degradation profile to the respective initial networks. It is noteworthy that the multi-stage mass loss can also be ascribed to the random scission of vinylic ends, head-to-head linkages and polymer backbones in the non-uniform co-

polymeric initial network prepared by conventional free radical polymerization[59].

The β-sheets-incorporated networks also exhibited remarkable stability at harsh pH conditions (2 M HCl and 2 M NaOH solution) and in the presence of protein denaturants (6 M GdnCl) (Fig. 8). After 4 weeks, the initial networks (hydrogels or cryogels) shrank or fell apart in strongly acidic or basic solutions whereas the β-sheets-rich networks remained intact. As revealed by mass loss study, a much more rapid loss was seen for the initial networks at harsh pH, due to the accelerated hydrolysis of ester groups of acrylate structures, while only a slight loss found for the β-sheets-rich networks under all these severe conditions, since the network architecture was stabilised by the non-covalent interactions of

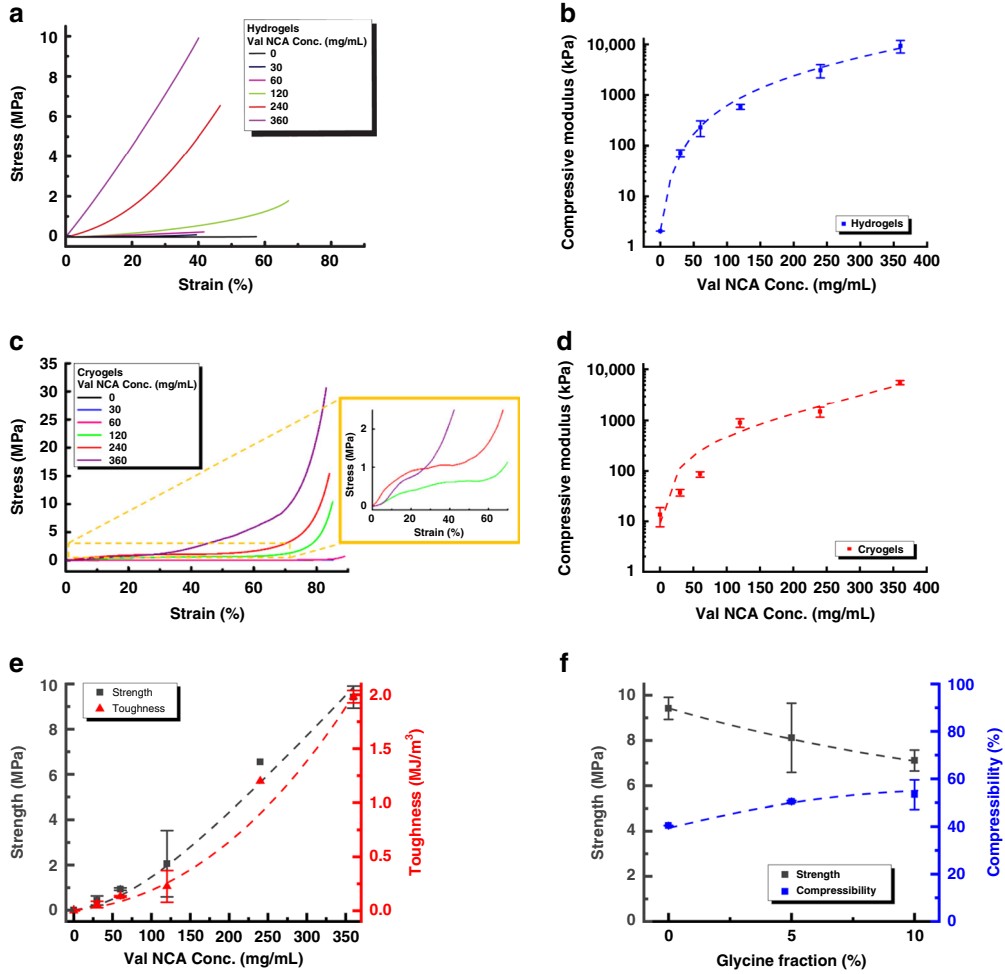

**Fig. 6 Mechanical properties summary of β-sheets-incorporated networks.** Networks obtained using different Val NCA monomer concentrations (0–360 mg/mL): **a**, **c** Compression strain-stress curves of hydrogels and cryogels respectively; **b**, **d** Compression moduli (stiffness) of hydrogels and cryogels respectively, as a function of Val NCA monomer concentrations; **e** Strength and toughness of hydrogels; **f** Strength and compression at break (compressibility) of hydrogels with glycine inclusion. All error bars represent standard deviation (*n* = 3).

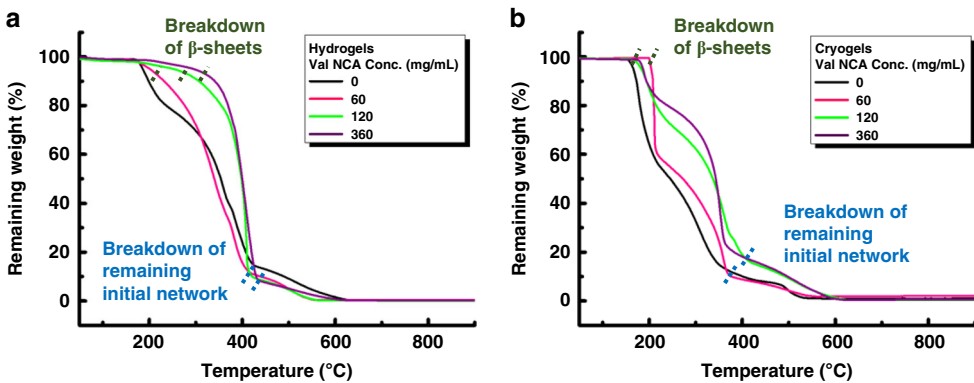

**Fig. 7 TGA analysis of the initial and β-sheet incorporated networks.** Networks made using different Val NCA concentrations (60, 120, 360 mg/mL): **a** hydrogels; **b** cryogels.

interpenetrated β-sheet clusters and the absence of ionisable groups also prevents dissociation of this secondary structure.

**Application within 3D-printed structures.** Recently, polypeptides synthesised via NCA ROP have been used as integral components of 3D-printed scaffolds[60–62]. Using our grafting-from strategy, β-sheet regions can be loaded into 3D-printed networks, endowing them with the enhanced physical properties defined above (Fig. 9). The initial 3D-printed structure was formed using a similar formulation to the initial networks stated before with an increased solids concentration and crosslinker-to-overall monomer ratio to form a stable enough 3D-printed initial network to handle (Fig. 9c, d). As pendant

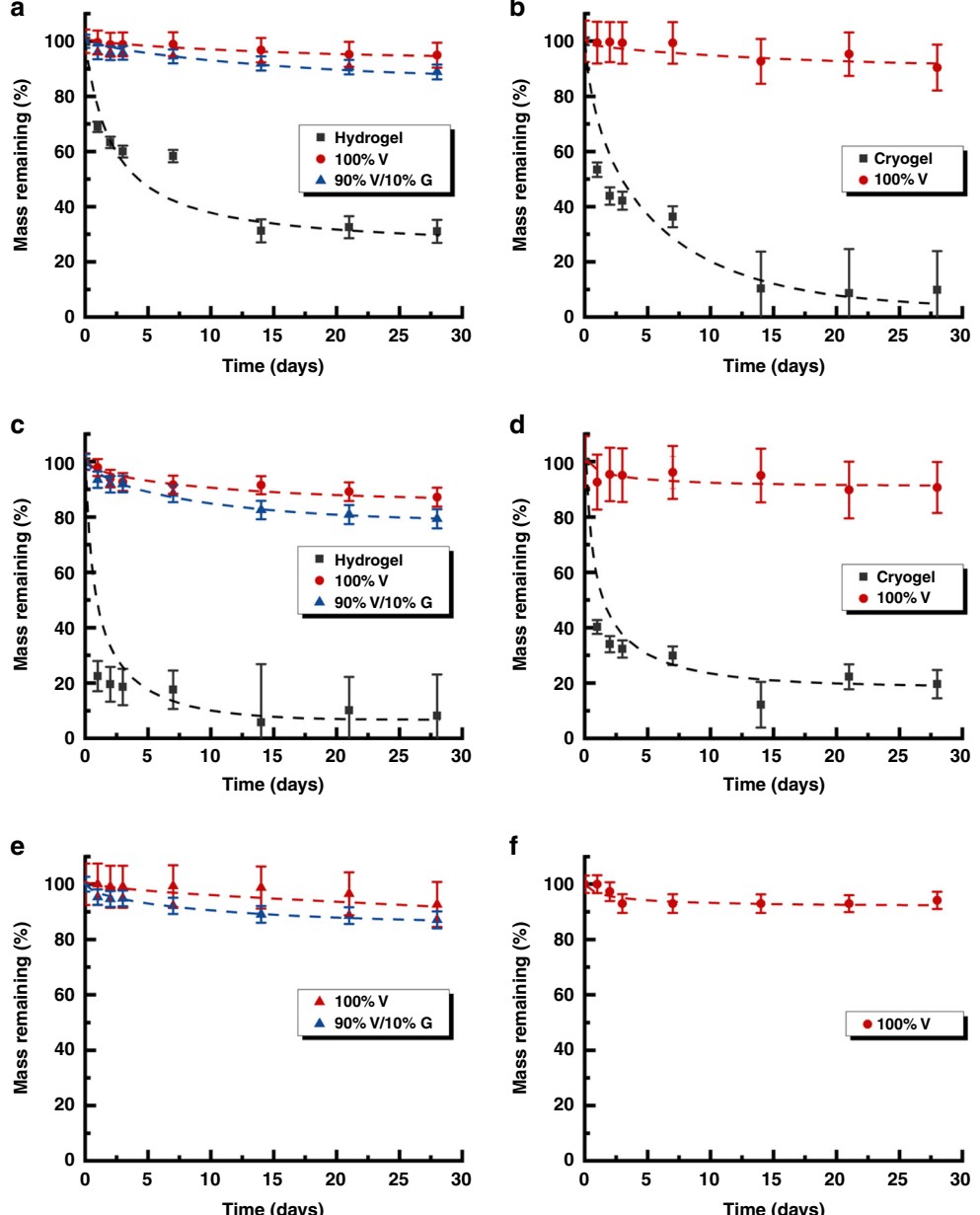

**Fig. 8 Degradation profiles of β-sheets-incorporated networks in harsh environments. a**, **c**, **e** Represent hydrogel degradation while **b**, **d**, **f** represents cryogels degradation. **a**, **b** Were obtained in 2 M HCl; **c**, **d** were obtained in 2 M NaOH and (**e**, **f**) were obtained in 6 M GdnCl. Error bars determined by experimental error.

amines are still present in the backbone of the scaffold, the network can be easily impregnated, albeit without the noticeable change in opacity observed in the other networks (Fig. 9c, f). While the networks did not show the same obvious visual change, the networks were noticeable stiffer and stronger when handling. To confirm that NCA ROP had occurred, FTIR spectroscopy was used to compare the network before and after NCA ROP (Fig. 9g). The amide A and B bands assigned to sharp peaks at around 3280 and 3085 cm$^{-1}$ are only present in the networks after NCA ROP has occurred as similarly observed in the networks that were not 3D-printed (shown in Fig. 5a). The presence of these amide stretches is only possible if NCA ROP has been successfully performed with this result showing the versatility of this technique and its potential in the 3D printing space.

**Discussion**

In conclusion, a strategy was developed to spatially control the growth of β-sheet forming polypeptides by using a soft polymeric network as an amorphous matrix to embed β-sheet nanocrystals within. The introduction of pendant amines into the polymer backbone of both hydrogels and cryogels to initiate guided NCA ROP allows for these β-sheets domains to adopt a conformation which enables these nanocrystals to be utilised effectively. This approach resulted in long chain poly(valine) grafts of up to 79 repeat units on average which would proceed to form a continuous β-sheet nanocrystal network. Subsequently, compressive strength was found to increase by 3 orders of magnitude up to 9.9 MPa for the hydrogels and 2 orders of magnitude up to 30 MPa for the cryogels. Furthermore, the properties of these networks can be easily tuned using glycine, with poly(valine-r-glycine)

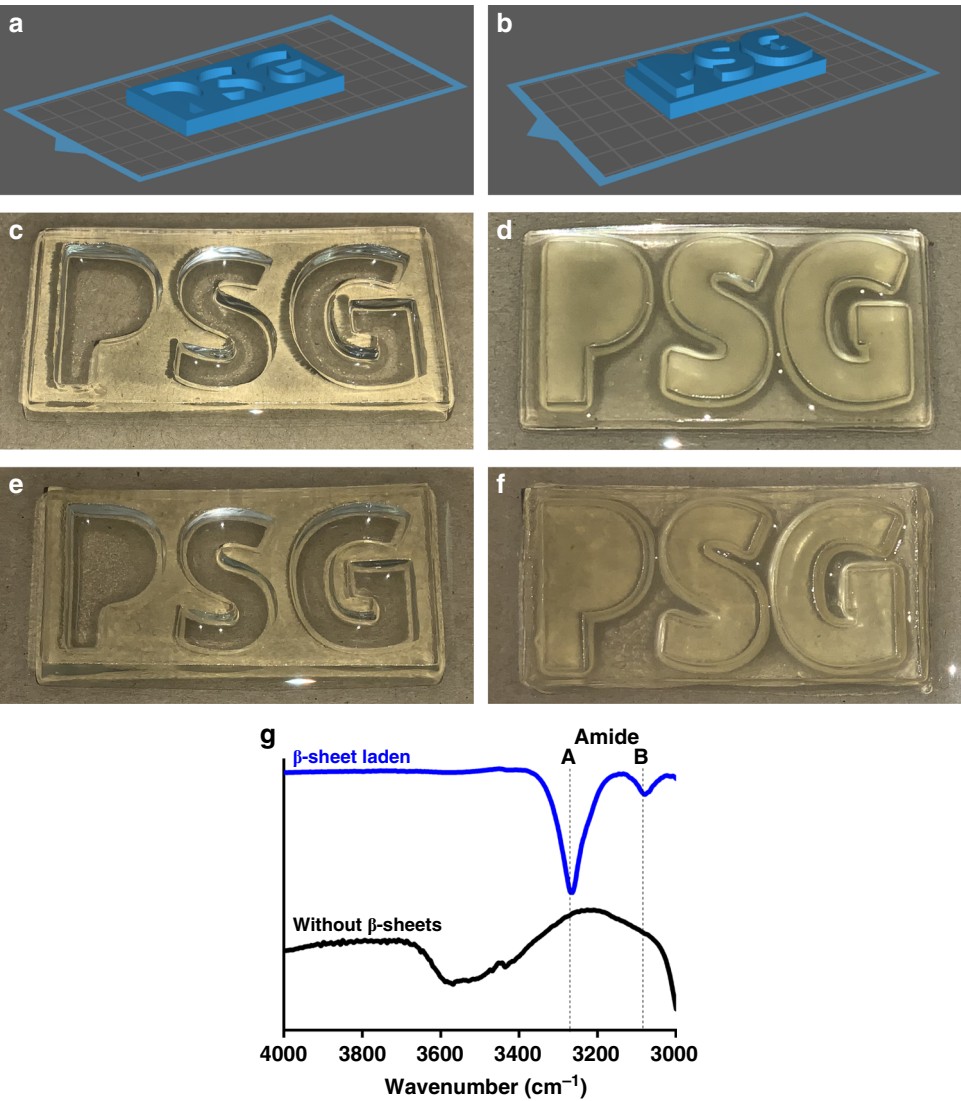

**Fig. 9 3D-printed networks with similar chemical composition to previous hydrogels. a, c, e** Are all based off a model printed with letters imprinted into a rectangular plate while **b, d, f** are all based off a model printed with letters protruding from a rectangular plate. **a, b** Are the original 3D models that were printed, **c, d** are the 3D-printed networks immediately after printing and **e, f** are the 3D-printed networks after NCA ROP had been performed on them. **g** ATR-FTIR of a 3D-printed network before and after NCA printing between 3000 and 4000 cm⁻¹.

grafts introducing an increased compressibility from 40 to 60% without a significant loss in toughness. This easily employable approach provides a powerful tool in the preparation of unique hybrid networks furnished with β-sheets and can be further used to tune the mechanical and degradation properties in other hydrogels for biomedical functionality and materials science.

## Methods

**Materials**. L-valine (≥98%) and glycine (≥98%) were purchased from Merck and used as received. Triphosgene (98%), (+)-α-pinene (99%), anhydrous n-pentane (>99%), anhydrous N,N-Dimethylformamide (DMF) (>99.8%), 2-aminoethyl methacrylate hydrochloride (97%), triethylamine (≥99.5%, TEA), oligo(ethylene glycol) methylether methacrylate (Mₙ = 1100, OEGMEMA), oligo(ethylene glycol) dimethacrylate (Mₙ = 750, OEGDMA), ammonium persulfate (APS, > 98%), N, N, N', N'-Tetramethyl ethylenediamine (TEMED, ≥ 99.5%), Diphenyl(2,4,6-trimethylbenzoyl)phosphine oxide (TPO, 97%) Thioflavin T (ThT), Guanidine hydrochloride (GdnCl, ≥99%), phosphate buffered saline (PBS) tablets, protease from streptomyces griseus (Type XIV), N-(3-Dimethylaminopropyl)-N'-ethyl carbodiimide (EDCl), 4-(Dimethylamino)pyridine (DMAP, ≥ 99%) and N, N, N', N", N"-pentamethyl diethylenetriamine (PMDETA, 99%) were all purchased from Sigma Aldrich and used as received. Anhydrous and deoxygenated Tetrahydrofuran (THF) (Honeywell, 99.9%, HPLC grade) was obtained by distillation from benzophenone and sodium metal under argon. AR grade hydrochloric acid

(HCl), sodium hydroxide (NaOH) and other solvents were purchased from ChemSupply Pty. Ltd. and used without further purification. L-valine (≥98%) was purchased from Merck and used as received. Anhydrous and deoxygenated Tetrahydrofuran (THF) (Honeywell, 99.9%, HPLC grade) was obtained by distillation from benzophenone and sodium metal under argon. Deuterated chloroform (CDCl₃) was purchased from Cambridge Isotope Laboratories and used as received. 2-(((butylthio)carbonothiolyl)thio)propanoic acid (TTC-1) was received from Dulux Group Australia and used as received. Methyl methacrylate (MMA, 99%, Sigma Aldrich) was de-inhibited by basic alumina prior to use.

**Synthesis of L-valine NCA (Val NCA)**. L-valine NCA was synthesised using a modified procedure from previously reported literature[44]. Briefly, L-valine (43 mmol, 5 g) was suspended within 80 mL of anhydrous THF in a two-necked round bottom flask under argon. Triphosgene (1.2× excess: 17 mmol, 5.1 g) was added and the mixture was continuously stirred 60 °C for 2 h or until all valine had dissolved. The reaction mixture was sparged with argon into a saturated NaOH solution for 1 h after which solvent was removed in vacuo (20 mbar, 40 °C) until equilibrium. The reduced mixture was redissolved in anhydrous 50 mL ethyl acetate which was chilled in 5 °C and then washed with saturated brine at 5 °C and 0.5 w/v % sodium bicarbonate aqueous solution at 5 °C with the organic phase being washed after separation each time. The resulting organic phase was dried using magnesium sulphate with the filtrate being reduced in vacuo (20 mbar, 50 °C) until equilibrium. The residue was recrystallised using n-pentane overnight. The resulting crystals were filtered and reprecipitated to afford a white powder (~60%

Yield). [1]H NMR (400 MHz, CDCl$_3$): δ$_H$ 0.92 (dd, 6 H, CH$_3$), 2.00–2.18 (m, 1 H, CH), 4.08 (dd, 1 H, cyclic CH), 8.84 (s, 1 H, cyclic NH).

**Synthesis of glycine NCA (Gly NCA)**. Glycine NCA was synthesised using a modified procedure from previously reported literature[63]. Glycine (67 mmol, 5 g) was suspended within a mixture of 250 mL of anhydrous THF and 21.2 mL (+)-α-pinene (133 mmol) in a two-necked round bottom flask under argon. Triphosgene (1.2× excess: 27 mmol, 7.9 g) was dissolved and added dropwise over 1.5 h while mixture was continuously stirred 70 °C and stirred for a further 1 h. The reaction mixture was sparged with argon into a saturated NaOH solution for 1 h after which the suspension was filtered and the filtrate was reduced in vacuo (20 mbar, 60 °C) until equilibrium. The resulting suspension was redissolved in THF and recrystallized using n-pentane overnight. The supernatant was removed and recrystallization was repeated two more times. The resulting white solid was filtered and washed with n-pentane (Yield 1.83 g, 27.2%). [1]H NMR (400 MHz, DMSO): δ$_H$ 4.33 (s, 2 H, cyclic CH$_2$), 8.84 (s, 1 H, cyclic NH).

**Synthesis of initial hydrogel network**. 2-aminoethyl methacrylate hydrochloride (1.56 mmol, 258 mg) was first deprotonated with TEA (1.88 mmol, ~300 μL) in 20 mL of DI water overnight. The mixture was mixed with co-monomer OEGMEMA (1.04 mmol, 1.14 g) and crosslinker OEGDMA (0.87 mmol, 650 mg) and degassed with N$_2$ for a few minutes. After addition of APS (20 mg) and TEMED (25 μL) (solids content ~10 w/w %, APS and TEMED 1–1.2 w/w% of solids), the solution was sonicated and separated into 1 mL batches in 3 mL syringes and allowed to stand at room temperature for 3 days. The resulting gels were swelled in DI water and washed six times to remove unreacted reactant. They were dehydrated in vacuo (20 mbar, 30 °C) for 2 days and stored in vacuo at room temperature for further use.

**Synthesis of initial cryogel networks**. The gel precursor solution was prepared and transferred to the syringes in the same procedures as described above. Then the solution in the mould was frozen at −18 °C and kept at this temperature for 3 days. After thawing and being washed with water, the cryogel sample was dried and stored in vacuo for further use.

**Synthesis of β-sheets-incorporated networks**. The following synthesis of HB360 is used as an example of the formation of β-sheet incorporated networks (refer to Table 1 for full experimental conditions). Dehydrated initial hydrogel network was immersed in the anhydrous DMF (3 mL) for 2 days to achieve swelling equilibrium. The swollen network was transferred to a monomer solution (in 3 mL anhydrous DMF) containing 360 mg/mL of Val NCA. The reaction vials were sealed at room temperature under vacuum for 3 days. the network was washed with DMF (3 × 3 mL) and sonicated in DMF (3 mL) for 30 min. To further remove potential non-grafted monomers/polymers, the network was then washed with THF (6 × 3 mL).

**Synthesis of PMMA-grated networks**. The initial network hydrogel or cryogel was swollen in 3 mL of TTC-1 (RAFT agent) (~40 mg/mL, 0.503 mmol, 6.5 equiv. of amine groups in networks) dissolved in anhydrous DMF. EDCI (90 mg, 0.468 mmol, 6 equiv.) and DMAP (1 mg, 0.0078 mmol, 0.1 equiv.) was then added in the mixture with the reaction sealed at room temperature with occasional agitation under vacuum for 2 days to the RAFT agent. Afterwards, the network was washed with anhydrous DMF for several times to remove the free RAFT agent and remaining catalyst until the network colour did not fade. The network attached with RAFT agent was then transferred to the MMA monomer solutions (in 3 mL anhydrous DMF) at different concentrations (60, 120, 360 mg/mL) in presence of PMDETA (~5 μL, 0.024 mmol, 0.33 equiv.). The reaction mixture was sealed and degassed using nitrogen for approximately 1 h. The UV light source ('Beaufly-nail lamp', 220 V, λ$_{max}$ ~ 365 nm, 4 × 9 W) was then switched on, and the reaction mixture was left at room temperature with occasional agitation under positive N$_2$ pressure for 3 days. A small aliquot (~100 μL) of the supernatant was taken for 1 H NMR analysis to check monomer conversion rate, according to an established calibration curve as described below. After the reaction, the supernatant was removed, and the network was washed with DMF (3 × 3 mL) and sonicated in DMF (3 mL) for 30 min. To further remove potential non-grafted monomers/polymers, the network was then washed with THF (6 × 3 mL). Finally, the network was dried and stored under vacuum.

**Synthesis of 3D-printed initial networks**. 2-Aminoethyl methacrylate hydrochloride (6.00 mmol, 994 mg), co-monomer OEGMEMA (4.00 mmol, 4.40 g) and crosslinker OEGDMA (10.0 mmol, 7500 mg) were dissolved in 25.8 mL of DMF: deionised water (7:3), with TPO photoinitiator (129 mg, 1 w/w% of solids) dissolved into the solution just prior to printing. The 3D printing was performed on an ANYCUBIC Photon 3D Printer with a λ$_{max}$ ~405 nm and a layer height of 0.5 mm, bottom layer exposure time of 120 s, bottom layer count of 8 and layer exposure time of 80 s. 3D Models were designed using Autodesk® Tinkercad and

sliced using ChiTuBox 3D Slicer software. The subsequent networks were then soaked in a solution of DMF (10 mL) and TEA (600 μL) overnight and then washed with anhydrous DMF (6 × 10 mL).

**Synthesis of β-sheets-incorporated 3D-printed networks**. The swollen networks were transferred to vials containing 50 mL of valine NCA in anh. DMF (120 mg/mL). The vials were then sealed and agitated at room temperature under vacuum for 3 days. The resulting networks were then washed with six aliquots of DI water, followed by subsequent washing with DMF (3 × 3 mL) before being sonicated in DMF (3 mL) for 30 min.

**Instrumentation**. Proton ([1]H) Nuclear Magnetic Resonance (NMR) spectroscopic analysis was performed on a Varian Unity Plus 400 MHz spectrometer using deuterated chloroform (CDCl$_3$) as the solvent. Attenuated total reflection- Fourier transform infrared spectroscopy (ATR-FTIR) spectroscopy was carried out using a Bruker Tensor 27 FTIR, with GladiATR ATR attachment obtained from Pike Technologies. The FTIR was equipped with OPUS 6.5 spectroscopy software from Bruker Optik GmBH. X-ray diffraction (XRD) spectroscopy was carried out on a Bruker D8 Advance Diffractometer, using standard Ni-filtered Cu kα radiation. Fluorescent images of ThT stained gel networks were acquired on a Leica TCS SP2 confocal laser scanning microscope (CLSM), using excitation wavelength of 490 nm. Scanning electron microscope (SEM) images were acquired using a FEI Quanta 200 ESEM FEG. Samples were pre-coated with gold using a Dynavac Mini Sputter Coater prior to imaging. Thermogravimetric analysis (TGA) was performed on a TGA/SDTA851e, Mettler Toledo in air with heating rate of 10 °C·min$^{-1}$.

**Determination of conversion within networks**. A calibration curve was generated from [1]H NMR analysis of Val NCA solutions (100 μL) at 5 different concentrations (15, 30, 60, 120, 180 mg/mL) diluted with CDCl$_3$ (900 μL). Integration of the Val NCA doublet peak at 0.92 ppm relative to the methyl DMF signals at 2.88 ppm and 2.96 ppm determined for the 5 Val NCA concentrations, afforded a calibration curve as shown in Supplementary Fig. 6.

Thus, Val NCA conversions were determined by taking 100 μL aliquots of reaction solutions after β-sheet incorporation into the networks and diluting with CDCl$_3$ (900 μL). These solutions were analysed by [1]H NMR by normalising against the same peaks of DMF, to determine Val NCA concentrations of each solution. This was then used to calculate the monomer conversion (as shown in Supplementary Fig. 7).

Gly NCA conversions were determined by [1]H NMR integration of the cyclic amine peak at 8.84 ppm relative to the Val NCA doublet at 0.92 ppm. Thus, the amount of NCA in solution due to Gly NCA and not Val NCA could be determined (see Supplementary Fig. 8).

**Swell ratio measurement**. To measure the equilibrium swell ratio ($Q$), the network was dried under heat in vacuo (60 °C, 20 mbar) for 24 h and then swollen in water until it reached equilibrium swelling. The weight of dry and fully swollen samples was determined by analytical balance and denoted as $W_d$ and $W_s$, respectively. The equilibrium swell ratio ($Q$) is defined by Eq. (1).

$$Q = \frac{W_s}{W_d} \tag{1}$$

**Degradation studies**. Dried networks were weighed and placed into different solutions. The vials were capped and placed into a temperature controlled orbital shaker (37 °C, 100 rpm). The samples were removed from each vial at certain time point and washed in deionised water. Then the samples were dehydrated by soaking in ethanol for 1 h followed by drying in vacuo (60 °C, 20 mbar) for 24 h. Finally, the samples were weighed, and the mass values obtained were plotted against time to obtain the degradation profiles.

**Mechanical tests**. The compressive strain-stress curves were obtained using an Instron Microtester 5848 equipped with a 2 kN static load cell and Bluehill material testing software. cylindrical hydrated network samples (diameter: ~9 mm; height: ~8.5 mm) were compressed to their maximum strain between two parallel plates at a crosshead speed of 0.1 mm/second. Engineering stresses and strains were recorded.

## Data availability

All data that support the findings of this study are available in the Figshare repository at https://doi.org/10.26188/5e1ff5e18580a. Additional data for this article are available as a Supplementary Information file.

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

## Acknowledgements

This work was performed in part at the Materials Characterization and Fabrication Platform (MCFP) at the University of Melbourne. N.J.C thanks The University of Melbourne for providing Australian Government Research Training Program Scholarship (AGRTP). D.G. thanks The University of Melbourne for providing Melbourne International Research Scholarships (MIRS). The authors would also like to thank Dr. Ke Xie and Mr. Min Liu for assistance with SEM measurements. We are also grateful to the Particulate Fluids Processing Centre (PFPC) for infrastructure support.

## Author contributions

N.J.C., D.G., S.T. and Q.F. established the reaction conditions of hydrogels and cryogels. N.J.C. performed the work introducing a new monomer into the system. S.T. and D.G. proposed the characterisation framework. N.J.C. and D.G. prepared samples and characterised the networks. N.J.C. and T.G.P. established the reaction conditions for the 3D-printed networks and subsequent characterisation. G.G.Q. and A.J.O. both supervised the project. N.J.C., D.G., S.T., Q.F. and G.G.Q. wrote the manuscript.

## Competing interests

The authors declare no competing interests.
