## [Peer Review File · Nature Communications]

Reviewers' Comments:

Reviewer #1:

Remarks to the Author:

The authors report the results of their studies on the development of spider silk inspired polymeric networks. The chemistry is simple and scalable and is described clearly and concisely. The characterisation of the polymers is sound.

The abstract is ok.

The body text is ok.

The supplementary information is ok.

The experimental appears reproducible.

The images in the supplementary information are useful.

The figures are ok albeit in need of error bars.

The structures of the polymers are interesting and related to others in the literature (a few additional references are noted below for inclusion).

Edits:

I'm unsure what "network guided assembly" in the title means and this is not clarified in the body text.

I'm unsure what "extensibility orders of magnitude" are - perhaps the authors mean extension at break?

Throughout the experimental section the experimental number must be explicitly stated.

Table 1 needs error bars for standard deviation on average conversions and DPs.

Table 2 needs error bars for standard deviation throughout.

Figure 1: error bars need to be clearer - perhaps make them thicker.

Figure 4: please correct "strech" to read "stretch"

Figure 5b, 5d, 5e, and 5f need error bars.

Figure 7 needs error bars.

Figure S4b and S4d need error bars.

References:

The authors have missed an important review on this class of materials:

1) Silk-inspired polymers and proteins. *Biochemical Society Transactions*. 2009, 37 (4), 677-681. DOI: 10.1042/BST0370677.

These other articles are related in terms of reinforced double network gels:

2) Grafting Techniques towards Production of Peptide-Tethered Hydrogels, a Novel Class of Materials with Biomedical Interest. *Gels*. 2015; 1(2): 194-218. DOI: 10.3390/gels1020194

3) Fundamentals of double network hydrogels. DOI: 10.1039/C5TB00123D

4) Effect of Polymer Entanglement on the Toughening of Double Network Hydrogels. DOI: 10.1021/jp052419n

Reviewer #2:

Remarks to the Author:

The manuscript describes the modification of a chemical hydrogel with poly(amino acid) side-chains that adopt a β -sheet conformation, yielding a toughened material that is unresponsive to

acid/base/protein denaturants. Whilst the generation of hydrogels that feature both physical crosslinks through β -sheet interactions, and covalent crosslinks is not novel (Org. Biomol. Chem., 2015, 13, 1983-1987, Angew. Chem. Int. Ed. 2011, 50, 8384 –8386), there is originality in the method of hydrogel preparation. I am not aware of research that deploys a chemical hydrogel as an initiator for N-carboxyanhydride ring-opening polymerisation, particularly for the creation of such materials. However, this may render the paper more suited to a materials/polymer chemistry journal. The case for publication would be strengthened if the impact of the materials could be explicitly shown, particularly in terms of applications that the general chemistry community can readily engage with, such as scaffolds for tissue engineering/self-healing coatings etc.

The manuscript is well written and largely scientifically sound. It also contains some nice features, such as including glycine within the formulation as a neat way to alter the mechanical properties of the hydrogels. However, I question the appropriateness of the research for publication in Nature Communications owing to its somewhat restricted impact/originality, and think that it is more suited to publication in a more specialised journal, unless appropriate material applications can be identified/shown.

Some comments that the authors should address:

- If the networks are inspired by spider silk, why is valine used to provide crystalline regions rather than alanine? The latter is a major component of fibroin.

- The abstract states 'The resultant continuous β -sheet nanocrystal network exhibited improved compressive strength and stiffness of up to 30 MPa (300 times greater) and 6 MPa (100 times greater) respectively.' 300 times greater than what? Presumably the chemical hydrogel but this should be stated.

- Scheme 1 is a little confusing. NH₂ is not written in the red circles (essential for NCA ROP) in the top box centre, and there is not a key that explains what the red arrows are in the top box right (polyvaline?)

- Further experimental detail to describe how NCA conversion was determined should be provided. ¹H NMR spectra that show how the integral ratio of NCA monomer (which peak?) and the solvent peak was used to determine NCA conversion should be included in the supporting information, along with the calibration curve used and all other NMR spectra.

- It is stated in the Crystallinity and parallel/antiparallel nature of β -sheets section that '.....in the initial network is not present in the other networks, indicating that all pendant amine groups had polypeptide chains successfully grafted from them.' However, the polypeptides would be expected to have terminal primary amine groups. Why is this not peak present post polymerisation?

- The FTIR data is described well in the text, but the image is not overly clear. The resolution should be enhanced, and I think that there should be a greater distance between each spectrum.

- In the Increased stability of networks under extreme conditions section it is stated '....more β -sheets start to degrade at a bit higher temperature.' Values for this temperature should be given (rather than referring to it as a bit higher). In addition, it is stated 'A thermal degradation process with several stages can be observed for these networks, implying the hybrid structure with different phases (e.g. crystalline β -sheet domains and amorphous phase of the initial network).' All these thermal transitions should be highlighted on the thermogram produced.

Comments pertaining to the literature

We would like to thank reviewer 1 and 2 for their comments on this manuscript. Below are the responses to the suggestions.

Reviewer #1:

I'm unsure what "network guided assembly" in the title means and this is not clarified in the body text.

Thank you for the comment. In response, we have clarified this within the body of the text in (i) the third paragraph of the Introduction, using the phrase "a pre-fabricated three-dimensional hydrophilic network which acts as a template to guide β -sheet assembly" and (ii) in the first paragraph of the Results, using the clause "The amine groups acted as initiating sites for NCA ROP of β -sheet forming polypeptides, resulting in a polymeric network which guides β -sheet assembly".

I'm unsure what "extensibility orders of magnitude" are - perhaps the authors mean extension at break?

In line with this recommendation, we have replaced the phrase "extensibility orders of magnitude" with "extension at failure".

Throughout the experimental section the experimental number must be explicitly stated.

Specific experimental numbers are inconsistent with the format of Nature Communications papers. However experimental codes have been included in the "Synthesis of β -sheets incorporated networks" subsection of the Experimental section to aid clarity including the following changes:

The subsection is prefaced with the following lines – "This is an example protocol for the synthesis of HB360. Please see Table 2 and 3 for information on monomer solution for soaking"

Protocol is specific to HB360 network, where the clause "...with differing ratios of glycine and valine NCA at different concentrations (30, 60, 120, 240, 360 mg/mL)" has been replaced with "...containing 360 mg/mL of Val NCA".

Furthermore, these experimental codes have been included as necessary within the main manuscript including the following changes:

"Synthesis of β -sheet rich networks through a grafting-from approach" subsection, paragraph 3 – clause "equivalent molar concentration to 360 mg/mL Val NCA solution" has been changed to "equivalent molar monomer concentration to HB360 and CB360"

Table 1 needs error bars for standard deviation on average conversions and DPs.

Thank you for the comment. We have included the error bars of the noted variables.

Experimental Code	Val NCA Conc. (mg/mL, mmol/mL)	Val NCA (mg, mmol)	Type of initial network	Conv. ^a (%)	Average DP ^b
HB30	30, 0.21	90, 0.62	Hydrogel	92.6 ± 8.9	7.47 ± 0.07
CB30			Cryogel	94.3 ± 4.9	7.60 ± 0.12
HB60	60, 0.42	180, 1.26	Hydrogel	88.6 ± 4.2	14.3 ± 0.26
CB60			Cryogel	91.2 ± 2.6	14.7 ± 0.45
HB120	120, 0.83	360, 2.52	Hydrogel	87.2 ± 6.7	28.1 ± 0.33
CB120			Cryogel	87.2 ± 8.2	28.1 ± 0.27
HB240	240, 1.67	720, 5.03	Hydrogel	83.4 ± 4.2	53.8 ± 1.00
CB240			Cryogel	77.2 ± 12.2	49.8 ± 0.32
HB360	360, 2.51	1080, 7.55	Hydrogel	80.1 ± 0.86	77.5 ± 7.04
CB360			Cryogel	74.5 ± 1.3	72.1 ± 4.37

Table 2 needs error bars for standard deviation throughout.

Thank you for the comment. We have included the error bars of the noted variables.

Experimental Code	Percentage of glycine (mol %)	Val NCA Conc. (mg/mL, mmol/mL)	Gly NCA (mg/mL, mmol/mL)	Val Conv. (%)	Gly Conv. (%)	Average DP (Val)	Average DP (Gly)
GH5	5	342, 2.39	12.7, 0.12	82.7 ± 8.3	87.3 ± 7.6	76.0 ± 0.72	4.2 ± 0.04
GH10	10	324, 2.26	25.4, 0.25	82.9 ± 11.9	88.3 ± 10.7	72.2 ± 0.47	8.5 ± 0.06

Figure 1: error bars need to be clearer - perhaps make them thicker.

Error bars have now been made more clear.

Figure 4: please correct "strech" to read "stretch"

Thank you for the correction, this error has been rectified.

Figure 5b, 5d, 5e, and 5f need error bars.

Thank you for the comment. We have included the error bars for the noted graphs.

Figure 7 needs error bars.

The graphs now have error bars included based on experimental error.

Figure S4b and S4d need error bars.

The graphs now have error bars included based on experimental error.

References:

After reading these references, we agree with the recommendation of including each of these references.

The authors have missed an important review on this class of materials:
 1) Silk-inspired polymers and proteins. *Biochemical Society Transactions*. 2009, 37 (4), 677-681. DOI: 10.1042/BST0370677.

This paper is now Reference 15 within the manuscript.

These other articles are related in terms of reinforced double network gels:
 2) Grafting Techniques towards Production of Peptide-Tethered Hydrogels, a Novel Class of Materials with Biomedical Interest. *Gels*. 2015; 1(2): 194–218. DOI: 10.3390/gels1020194

This paper is now Reference 41 within the manuscript.

3) Fundamentals of double network hydrogels. DOI: 10.1039/C5TB00123D
 4) Effect of Polymer Entanglement on the Toughening of Double Network Hydrogels. DOI: 10.1021/jp052419n

These papers are now Reference 52 and 53 respectively within the manuscript.

Reviewer #2 (Remarks to the Author):

The manuscript is well written and largely scientifically sound. It also contains some nice features, such as including glycine within the formulation as a neat way to alter the mechanical properties of the hydrogels. However, I question the appropriateness of the research for publication in Nature Communications owing to its somewhat restricted impact/originality and think that it is more suited to publication in a more specialised journal, unless appropriate material applications can be identified/shown.

To demonstrate the impact of this approach as a broad methodology, we have created a proof of concept for the use of this strategy within 3D printed networks. As a result, we have included a subsection in our Results named “Application within 3D printed structures” which stipulates this concept.

Some comments that the authors should address:

- If the networks are inspired by spider silk, why is valine used to provide crystalline regions rather than alanine? The latter is a major component of fibroin.

Thank you for this comment. Our attempts at mimicry centre around spatial placement of β -sheets within an amorphous network instead of the specific residues involved show a proof of concept. As alanine is well known to favour α -helices, we have opted to use valine as a well-known β -sheet forming block. This has been clarified in the following statements:

Intro paragraph 4: “...success with NCA ROP” to “...success with NCA ROP (specifically valine NCA ROP)”

Intro paragraph 5: Added sentence “It should be noted that while natural spider-silk uses blocks of alanine to form β -sheets, they actually favour α -helical conformations and require specific environmental condition or blocks must remain in a low DP to instead form the desired β -sheets. Instead, valine blocks were employed due to our group’s previous experience with it and its high propensity to form β -sheets.”

- The abstract states ‘The resultant continuous β -sheet nanocrystal network exhibited improved compressive strength and stiffness of up to 30 MPa (300 times greater) and 6 MPa (100 times greater) respectively.’ 300 times greater than what? Presumably the chemical hydrogel but this should be stated.

Thank you for noting the lack of clarity. We have included the phrase “over the initial network lacking β -sheets” and referred to these mechanical properties as have qualities of “30 MPa (300 times greater than the initial network) and 6 MPa (100 times greater than the initial network) respectively”

- Scheme 1 is a little confusing. NH₂ is not written in the red circles (essential for NCA ROP)

in the top box centre, and there is not a key that explains what the red arrows are in the top box right (polyvaline?)

Thank you for the comment. We have modified Scheme 1 by moving the primary amine as an indicator to outside of the representative molecule such that it is not required within the red circles. Furthermore, we have included an indication of the chemical structure of the polypeptides and appropriately associated it to the red arrows.

- Further experimental detail to describe how NCA conversion was determined should be provided. ^1H NMR spectra that show how the integral ratio of NCA monomer (which peak?) and the solvent peak was used to determine NCA conversion should be included in the supporting information, along with the calibration curve used and all other NMR spectra.

We have stipulated the method used to determine conversion in the experimental section titled “Determination of conversion within networks” and included the NMR spectra, calibration curves and calculations in the Supporting Information (Figures S5-7).

- It is stated in the Crystallinity and parallel/antiparallel nature of β -sheets section that ‘.....in the initial network is not present in the other networks, indicating that all pendant amine groups had polypeptide chains successfully grafted from them.’ However, the polypeptides would be expected to have terminal primary amine groups. Why is this not peak present post polymerisation?

Upon closer examination of the spectrum, we have concluded that a slight shoulder does indicate the presence of the aforementioned amines. We thank the reviewer for this clarification have included an inset in this figure and made the following corrections in response:

Replaced “As NCA ROP is initiated, the primary amine assigned to a broad peak between 3300 and 3500 cm^{-1} in the initial network is not present in the other networks, indicating that all pendant amine groups had polypeptide chains successfully grafted from them. The amide A

and B which are both associated with N-H stretching vibrations bonds have sharp bands at around 3280 and 3085 cm^{-1} assigned to them in all double bonded networks without glycine.⁴⁵ The amide A band is shown to have a small shoulder at 3300 cm^{-1} once glycine is introduced which is more evident in samples containing more significant amounts of glycine (Figure S3).⁴⁵ As the amide A bond is largely sensitive to hydrogen bonding strength, this can be associated to the change in bond strength.⁴⁵ with “As NCA ROP is initiated, the amide A and B bands assigned to sharp peaks at around 3280 and 3085 cm^{-1} respectively which are both associated with N-H stretching vibrations bonds are prominent in all networks except for the initial network, indicating successful polymerization.⁴⁵ The amide A band is shown to have a small shoulder at 3300 cm^{-1} once glycine is introduced which is more evident in samples containing more significant amounts of glycine (Figure S3).⁴⁵ As the amide A bond is largely sensitive to hydrogen bonding strength, this can be associated to the change in bond strength. The primary amine assigned to a broad peak between 3300 and 3500 cm^{-1} is found to overlap with the amide A peak in these networks while being clearly seen in the initial network, providing a shoulder for these peaks. The presence of this primary amine is to be expected as the terminal primary amine of polypeptides remains after NCA ROP though its relative strength is reduced with increasing β -sheet induction due to the increase of amide bonds. “

Removed: “(except for the NH_2 stretch)”

- The FTIR data is described well in the text, but the image is not overly clear. The resolution should be enhanced, and I think that there should be a greater distance between each spectrum.

In response, we have enhanced the resolution of Figure 4 and provided enough distance in between spectrums such that there is no chance of interference between spectrums.

- In the Increased stability of networks under extreme conditions section it is stated ‘...more β -sheets start to degrade at a bit higher temperature.’ Values for this temperature should be given (rather than referring to it as a bit higher). In addition, it is stated ‘A thermal degradation process with several stages can be observed for these networks, implying the hybrid structure with different phases (e.g. crystalline β -sheet domains and amorphous phase of the initial network).’ All these thermal transitions should be highlighted on the thermogram produced.

Thank you for the comment. We have changed the language to be more precise, stating that “ β -sheets start to degrade at higher temperatures, with the hydrogels synthesised from 360 mg/mL Val NCA solution displaying significant degradation at 160 °C greater than the initial network and cryogels synthesised from 360 mg/mL Val NCA solution displaying significant degradation at 40 °C greater than the initial network.” Furthermore, language has been clarified to refer to the different stages of degradation including the line “Once the β -sheet network is completely degraded, the β -sheets incorporated networks follow a similar degradation profile to the respective initial networks.” and the thermogram has been labelled as below.

Reviewers' Comments:

Reviewer #1:

Remarks to the Author:

The authors report the results of their studies on the development of polymers with interesting mechanical properties. The manuscript is an interesting read, but in need of editing.

The abstract is ok.

The body text is ok, albeit in need of editing.

The level of experimental details is ok.

The images in the supplementary information are useful.

The figures are ok.

Edits:

The sentences "It should be noted that although natural spider-silk uses blocks of alanine to form β -sheets, they actually favour α -helical conformations. In order to form the desired β -sheets the silk depends on specific environmental conditions and contain blocks with low molecular weights." are an oversimplification. Please change them to read "Natural spider-silk uses blocks of alanine to form β -sheets during the natural silk fiber production process which involves specific environmental changes to encourage the formation of the β -sheets."

Please change "However, the synthetic preparation of mechanically strong β -sheet materials remains significantly challenging due to the insoluble assemblies formed through hydrophobic association of residues." to read "However, the preparation of mechanically strong β -sheet rich materials remains a significant challenge due to the challenges involved in processing the polymers/proteins and managing the assembly process of the hydrophobic residues."

The term "extension at failure" is incorrect - the correct terminology "extension at break" must be used.

Please delete the sentence "It should be noted that although natural spider-silk uses blocks of alanine to form β -sheets, they actually favour α -helical conformations." because this is not correct - one of the challenges in solid phase synthesis of alanine rich peptides is because of its tendency to fold into beta sheets on the solid support.

Please change "resulting in a polymeric network which guides β -sheet assembly" to read "resulting in a polymeric network incorporating non-covalent β -sheet crosslinks"

Please change "swelling ratio" to read "swell ratio" throughout the manuscript, including figures.

Please do not arbitrarily capitalise letters in sentences "Scanning Electron Microscopy (SEM)" should read "scanning electron microscopy (SEM)"

"Glycine NCA" yield of 12% is low - I'm surprised this was not optimised prior to submission.

Please delete "Supporting Information is available from the Wiley Online Library or from the author." this is an embarrassing oversight.

Figure 1: please give the diameter of the glass containers. Is the water at pH7? is it distilled, deionized, ultrapure?

Figure 4: "N-H2" should read "NH2"

Figure 5: what scale are the photos?

Supplementary information.

The figure legends do not need to be underlined - please delete the underlining.

Figure S5: NMRs should have their baselines corrected - particularly: b/c/d/e.

Label: S6 appears twice - please renumber S6, S7... and correct references in the main paper accordingly.

Reviewer #2:

Remarks to the Author:

The authors have provided a detailed response that satisfies the points that were raised in the initial review. Consequently, I support the publication of this manuscript.

Point by Point Reviewer Response

Reviewer #1:

Reviewer 1 commented: The sentences "It should be noted that although natural spider-silk uses blocks of alanine to form β -sheets, they actually favour α -helical conformations. In order to form the desired β -sheets the silk depends on specific environmental conditions and contain blocks with low molecular weights." are an oversimplification. Please change them to read "Natural spider-silk uses blocks of alanine to form β -sheets during the natural silk fiber production process which involves specific environmental changes to encourage the formation of the β -sheets."

We have included the text as per the reviewer's request on page 4

R1: Please change "However, the synthetic preparation of mechanically strong β -sheet materials remains significantly challenging due to the insoluble assemblies formed through hydrophobic association of residues." to read "However, the preparation of mechanically strong β -sheet rich materials remains a significant challenge due to the challenges involved in processing the polymers/proteins and managing the assembly process of the hydrophobic residues."

We have made this change as suggested on page 1.

The term "extension at failure" is incorrect - the correct terminology "extension at break" must be used.

We have changed the term "extension at failure" to "extension at break".

Please delete the sentence "It should be noted that although natural spider-silk uses blocks of alanine to form β -sheets, they actually favour α -helical conformations." because this is not correct - one of the challenges in solid phase synthesis of alanine rich peptides is because of its tendency to fold into beta sheets on the solid support.

While this is true in the solid phase synthesis, this is due to the short length of the alanine blocks. As studied by multiple sources (Vanhalle, Corneillie ¹, Fujie, Kōmoto ², Yang and Honig ³), higher molecular alanine blocks do favour α -helices. As such, this sentence has been referenced and replaced with the following line for clarification: "although it is well known that alanine forms β -sheets at low molecular weight (small blocks), high molecular weight (larger blocks) alanine has been shown to form β -sheets."

Please change "resulting in a polymeric network which guides β -sheet assembly" to read "resulting in a polymeric network incorporating non-covalent β -sheet crosslinks"

We have modified the term per the reviewer's request.

Please change "swelling ratio" to read "swell ratio" throughout the manuscript, including figures.

We have modified the term in the main body, experimental section and Figure 1 as per the reviewer's request.

Please do not arbitrarily capitalise letters in sentences "Scanning Electron Microscopy (SEM)" should read "scanning electron microscopy (SEM)"

We have removed the capitalisation per the reviewer's request.

"Glycine NCA" yield of 12% is low - I'm surprised this was not optimised prior to submission.

We have used an alternate procedure which was found to have a greater yield with comparable results. As such, we have introduced the new protocol with greater yield (27.2%). The low yield is due to the difficulty associated to glycine NCA synthesis.

Please delete "Supporting Information is available from the Wiley Online Library or from the author."

We have deleted this section and included the line "Additional data for this article is available as a supplementary information file." under the Data Availability Statement as is consistent with other papers submitted to Nature Communications.

Figure 1: please give the diameter of the glass containers. Is the water at pH7? is it distilled, deionized, ultrapure?

The caption for Figure 1c) now reads "Photos showing the cryogels with different content of β -sheets in deionised water (glass vial diameter ~25 mm) at pH 6.7."

Figure 4: "N-H2" should read "NH2"

As FTIR specifically refers to the stretching vibrations of the covalent bonds, "N-H2" has been changed to "N-H stretch (primary amine)"

Figure 5: what scale are the photos?

For clarification scale bars have been added to Figure 3.

Supplementary information.

The figure legends do not need to be underlined - please delete the underlining.

Figure captions are no longer underlined.

Figure S5: NMRs should have their baselines corrected - particularly: b/c/d/e.

All NMRs have been baseline corrected. This had a minor effect on the values for Conv. and Average DP in Table 1 and Val Conv., Gly Conv. and Average DP (Val and Gly) in Table 2, which have all been changed as a result and the line “however, valine conversion remains similar to those with 100% valine.” has been changed to “while valine conversion increased to ~82-84%.” to reflect this change.

Label: S6 appears twice - please renumber S6, S7... and correct references in the main paper accordingly.

This label has been changed to S7 in the supplementary information. Figure S7 has already been corrected in the main paper.

Reviewer #2:

The authors have provided a detailed response that satisfies the points that were raised in the initial review. Consequently, I support the publication of this manuscript.

No modifications have been highlighted by this reviewer and as such no further changes were made.

References for Response

1. Vanhalle M, Corneillie S, Smet M, Van Puyvelde P, Goderis B. Poly(alanine): Structure and Stability of the d and l-Enantiomers. *Biomacromolecules* **17**, 183-191 (2016).
2. Fujie A, Kōmoto T, Ōya M, Kawai T. Crystallization of polypeptides in the course of polymerization, III. Further studies on the growth mechanisms of poly(L-alanine) crystals. *Die Makromolekulare Chemie* **169**, 301-321 (1973).
3. Yang A-S, Honig B. Free Energy Determinants of Secondary Structure Formation: I. α -Helices. *J Mol Biol* **252**, 351-365 (1995).

Reviewers' Comments:

Reviewer #1:

Remarks to the Author:

The authors report the results of their studies on the development of polymers with interesting mechanical properties. The manuscript has been revised in line with reviewers suggestions and is improved.

The abstract is ok.

The body text is ok.

The level of experimental details is ok.

The images in the supplementary information are useful.

The figures are ok.